# Sexual Dimorphism in Third Molar Agenesis in Humans with and without Agenesis of Other Teeth

**DOI:** 10.3390/biology11121725

**Published:** 2022-11-28

**Authors:** Ragda Alamoudi, Mohammed Ghamri, Ilias Mistakidis, Nikolaos Gkantidis

**Affiliations:** 1Department of Orthodontics and Dentofacial Orthopedics, School of Dental Medicine, University of Bern, 3010 Bern, Switzerland; 2Directorate of Health Affairs-Jeddah, Ministry of Health, Riyadh 11176, Saudi Arabia; 3Private Practice, 1441 ZN Purmerend, The Netherlands

**Keywords:** tooth agenesis, third molars, sexual dimorphism, prevalence, patterns, humans

## Abstract

**Simple Summary:**

The assessment of differences between sexes (sexual dimorphism) in tooth formation allows for a better understanding of the developmental processes that contribute to phenotypical variations. Here, we tested for sexual dimorphism in the absence of third molar formation (agenesis) in modern humans with and without agenesis of teeth other than the third molars. No sexual dimorphism was detected in the patterns or the severity of third molar agenesis in any group. When teeth other than the third molars were absent, both sexes showed third molar agenesis and bilateral occurrence more often. There was no sexual dimorphism in the patterns or the severity of third molar agenesis, despite the higher vulnerability compared to other teeth. Moreover, there was no sexual dimorphism in single tooth agenesis with regard to any tooth. This indicates that the third molars are more often and more globally affected by genetic or epigenetic factors involved in tooth agenesis, with no differences between sexes. This is consistent with the evolutionary trend in humans towards a reduced molar number. In preadolescent patients with multiple tooth agenesis, the higher possibility of additional third molar agenesis should be considered during dental treatment planning.

**Abstract:**

Sexual dimorphism in the human dentition is of interest from a developmental, evolutionary, and clinical point of view. Here, we investigated sexual dimorphism in third molar agenesis patterns and severity in non-syndromic white European individuals with (group A: 303 individuals) and without agenesis (group B: 303 individuals) of teeth other than the third molars. There was no sexual dimorphism in the patterns or the severity of third molar agenesis within groups. Both sexes showed a higher number of third molar agenesis per individual in group A than in group B. The most common third molar agenesis pattern was that of no third molars. For both females and males, bilateral third molar agenesis was approximately three times more frequent in group A than in group B (*p* < 0.001), whereas no difference was detected for unilateral agenesis. These findings indicate a strong genetic control of the developmental process of tooth formation, with any disruptions affecting both sexes in a similar manner. Overall, the higher vulnerability of third molar formation could be associated with the evolutionary trend in humans towards a reduced number of molar teeth, which seems to show no sex-related differences.

## 1. Introduction

Tooth agenesis is the most common congenital anomaly of the craniofacial area in humans [1,2]. Defined as the congenital absence of one or more teeth, tooth agenesis can be a phenotypic feature of a syndrome, such as ectodermal dysplasia, cleft lip, cleft palate, Down syndrome, and Van der Woude syndrome, or it can have an isolated form (non-syndromic tooth agenesis) [2]. About 22.6% of the population has at least one third molar agenesis [1], while agenesis of other permanent teeth, excluding third molars, is evident in 6.4% of the population [2]. The phenotypic expression of tooth agenesis varies considerably between individuals regardless of third molar agenesis [3,4].

Without considering the third molars, sexual dimorphism in permanent tooth agenesis prevalence was observed in various studies, with females showing higher prevalence [2]. Sexual dimorphism in the agenesis patterns of teeth other than the third molars has also been reported in the literature [5,6]. A recent study in Austria reported that non-syndromic males were less often affected by tooth agenesis, but showed more severe patterns. Males had significantly higher numbers of congenitally missing first premolars (apart from upper right first premolar), mandibular incisors (central and lateral), and maxillary left first permanent molars. On the other hand, agenesis of the upper right first premolar was more often evident in females [5]. Moreover, in a non-syndromic Japanese population with oligodontia, sexual dimorphism was evident in tooth agenesis patterns, but not in the distribution of the number of missing teeth. The agenesis prevalence of the maxillary second premolars was significantly higher in females and that of the mandibular central incisors was significantly higher in males [6]. To our knowledge, the aforementioned studies were the only ones testing sexual dimorphism in tooth agenesis patterns, and they excluded third molars.

According to a recent meta-analysis, females were 14% more likely than males to have agenesis of one or more third molars. Overall, maxillary third molar agenesis was 36% more likely than mandibular agenesis, and large deviations in the prevalence of third molar agenesis were evident between different geographic regions [1]. The aforementioned meta-analysis tested third molar agenesis as a binary outcome (yes/no) and did not investigate sexual dimorphism in third molar agenesis patterns or in relation to other missing teeth in the dentition.

Third molar formation has the highest variation in all teeth [1,2]. Third molars are the last teeth to develop in the dentition and show instability in terms of developmental integrity and timing [1,7,8]. So far, the few existing studies on sexual dimorphism in tooth agenesis patterns are focused on individuals with agenesis of other teeth, ignoring the third molars [5,6].

Thus, previous studies suggest sex-specific effects on the agenesis patterns of teeth other than third molars [5,6], and higher odds (14%) for females to exhibit agenesis of at least one third molar [1]. However, sexual dimorphism in third molar agenesis patterns and the association of agenesis on teeth other than third molars to third molar formation in different sexes remain largely unexplored. This information could enable a better understanding of the variation in the number of human teeth, both from evolutionary and developmental viewpoints [9]. A previous study revealed an association between the number of formed teeth and craniofacial shape, which suggested a broad relationship between number of teeth and overall craniofacial development, but no sexual dimorphism was detected in this association [10]. Moreover, it was found that modern individuals with tooth agenesis have smaller facial configurations [11], which is also evident when considering only the third molars [12]. Likewise, there was no significant mediation of the effects by sex factor in the aforementioned studies [11,12]. The investigation of sexual dimorphism in third molar agenesis patterns could provide further insight in the classical debate on whether the presence of third molars is controlled by the same genetic mechanisms as the other permanent teeth, or their formation follows a separate developmental pattern [13]. Based on previous findings on the number of teeth formed in a human dentition [1,2,4], we suspect that, in the presence of agenesis in teeth other than third molars, females, which show higher tooth agenesis incidence, might also have more severe third molar agenesis patterns. On the other hand, considering the association of number of teeth with craniofacial size and shape, where both sexes are similarly affected [10,11,12], this might not be the case.

Therefore, the present study aims to investigate sexual dimorphism in the patterns and the severity of third molar agenesis in a modern human group without agenesis of teeth other than third molars and compare to a group with agenesis of other teeth.

## 2. Materials and Methods

This study complies with the STROBE (Strengthening the Reporting of Observational Studies in Epidemiology) guidelines for presenting observational studies.

### 2.1. Sample

The present retrospective data analysis was performed on an already available sample [4] collected from orthodontic patient records, which were obtained within a 12-year period (2006–2018) at the following orthodontic clinics: (a) the University of Bern, Switzerland, (b) the University of Athens, Greece, (c) two private practices in Athens and two in Thessaloniki, Greece, and (d) one private practice in Biel, Switzerland.

At the time of sample collection, each participant was assigned a randomly generated code and all identification information was immediately removed. The sample collection aimed at forming the two following groups:

Group A (*n* = 303) included non-syndromic European (white) individuals, aged between 12.5 and 40 years, with agenesis of teeth other than the third molars. They did not have any systemic diseases or other defects that could affect craniofacial complex development, as reported in the subjects’ medical records, and they had adequate quality panoramic radiographs.

Group B (*n* = 303) included non-syndromic European (white) individuals without agenesis of teeth other than the third molars, who were matched for age (within 6 months) and sex to Group A. All other eligibility criteria for Group B were the same as those mentioned above for Group A.

The third molar formation was not considered during the original sample selection. Moreover, individuals with unknown or uncertain etiology of any missing teeth were excluded from the study.

A sample size calculation was not performed prior to the study implementation, since all individuals that comprised the pre-existing sample and fulfilled the inclusion criteria were included in the present study. Furthermore, a direct sample size calculation would have been difficult in the absence of preexisting data, as well as for a single primary outcome. The current approach resulted in a relatively large sample size, as indicated by empirical evidence and previous relevant publications [4,6,14,15].

### 2.2. Data Extraction

All patient files (medical and dental history, intraoral and extraoral photos, and radiographs) were reviewed, and all data were recorded in an Excel sheet (Microsoft Excel, Microsoft Corporation, Redmond WA, USA) in a standardized manner according to a previous protocol [4]. Data extraction was performed previously by a single researcher and was repeated for 40 randomly selected subjects at 1 month after the initial assessment (https://www.random.org/, accessed on 3 March 2020). Adequate intra-rater agreement was reported [4]. 

The following variables were also recorded: gender, date of birth, date of panoramic radiograph, and congenitally missing teeth, including third molars. All panoramic radiographs were viewed on a computer screen to identify tooth abnormalities. For this purpose, the printed radiographs were scanned using an Epson Perfection V700 scanner with a resolution of 600 dpi and a scale of 1:1 and saved in .tiff format (tagged image file format).

### 2.3. Tested Outcomes

The following outcomes were tested in this study:-Sexual dimorphism in the patterns of third molar agenesis within groups.-Sexual dimorphism in the severity of third molar agenesis within groups.-Sexual dimorphism in the patterns and the severity of third molar agenesis between groups.-Sexual dimorphism in the patterns of tooth agenesis in Group A.

### 2.4. Statistical Analysis

All statistical analyses were conducted using SPSS software (IBM SPSS Statistics for Windows, Version 28.0. IBM Corp, Armonk, NY, USA). A two-tailed Pearson’s Chi-square test was used to assess differences between the frequencies observed in the two groups, when more than 15 observations were available per group. In cases of fewer observations, the Fisher’s exact test was used instead. The Spearman’s correlation coefficient was used to investigate the relation between the number of agenesis of teeth other than the third molars and the number of third molar agenesis, overall as well as within quadrants.

The level of significance was set at 0.05. A Bonferroni correction was applied in the case of multiple pairwise tests for similar outcomes to avoid false-positive results.

## 3. Results

### 3.1. Sexual Dimorphism in Third Molar Agenesis Associated with Agenesis of Other Teeth (Group A)

From a potential total sample of 532 third molars in males and 680 in females (133 male and 170 female participants), there were 172 (32.3%) and 246 (36.2%) third molars missing, respectively (*p* > 0.01). There was no sexual dimorphism in third molar agenesis severity. The average number of third molar agenesis per individual was 1.29 ± 1.59 in males and 1.45 ± 1.60 in females (*p* > 0.01). There was no sexual dimorphism evident when comparisons were performed within jaws or quadrants (*p* > 0.006, Table 1, Appendix A). Furthermore, there were no differences within sexes in the type of third molar missing when comparisons were performed between or within jaws (*p* > 0.01, Figure 1, Appendix A).

In Group A, bilateral third molar agenesis was present statistically more often than unilateral agenesis in both sexes (*p* < 0.001). The most common third molar agenesis pattern was agenesis of all third molars (20.3% in males and 18.8% in females), followed by bilateral agenesis of lower third molars (6% in males and 6.5% in females) (Table 2). The incidence of bilateral third molar agenesis was 2.6 and 3.7 times higher than unilateral agenesis in males and females, respectively (*p* = 0.300, Table 3). As evident in Table 2, there was no sexual dimorphism in the bilateral versus the unilateral occurrence of the most common patterns.

There was no sexual dimorphism in the incidence of third molar agenesis when distinct groups were formed based on agenesis severity, considering all possible categories (0, 1, 2, 3, or 4 missing molars, Figure 2, Appendix A).

When all teeth were considered, the most common single tooth missing was the lower second premolar (38.7% in males and 38.5% in females), followed by the lower third molar (31.6% in males and 37.6% in females), the upper third molar (33% in males and 34.7% in females), and the upper lateral incisor (29.7% in males and 26.2% in females). For all single teeth, there was no difference in the distribution between right and left sides, as well as no sexual dimorphism (*p* > 0.001, Figure 3, Appendix A). The most common tooth agenesis patterns in the entire dentition, as well as in the maxilla and the mandible, are ranked for both sexes in Table 4.

### 3.2. Comparison of Sexual Dimorphism in Third Molar Agenesis between Group A and Group B

In Group B, which was matched for age and sex to Group A but included individuals without agenesis of any tooth apart from the third molars, 60 third molars were missing in males (prevalence 11.3%) and 84 in females (prevalence 12.4%) (*p* = 0.592). The average number of third molar agenesis per individual in Group B was 0.45 ± 1.10 in males and 0.49 ± 1.07 in females (*p* > 0.01). Similar to Group A, there was no sexual dimorphism in Group B when considering the distribution of third molar agenesis within jaws or quadrants (*p* > 0.006, Table 5, Appendix A). Furthermore, there were no differences within sexes in the type of third molar missing when comparisons were performed between or within jaws (*p* > 0.01, Figure 1, Appendix A).

In Group B, which was matched for age and sex to Group A but included individuals without agenesis of any tooth apart from the third molars, 60 third molars were missing in males (prevalence 11.3%) and 84 in females (prevalence 12.4%) (*p* = 0.592). The average number of third molar agenesis per individual in Group B was 0.45 ± 1.10 in males and 0.49 ± 1.07 in females (*p* > 0.01). Similar to Group A, there was no sexual dimorphism in Group B when considering the distribution of third molar agenesis within jaws or quadrants (*p* > 0.006, Table 5, Appendix A). Furthermore, there were no differences within sexes in the type of third molar missing when comparisons were performed between or within jaws (*p* > 0.01, Figure 1, Appendix A).

The incidence of third molar agenesis in Group A (females: 52.4%, males: 48.9%) was significantly higher for both sexes than the incidence in Group B (females: 21.8%, males: 18.9%) (*p* < 0.001, Appendix A).

In contrast to Group A, there was no difference in Group B in the incidence of bilateral versus unilateral third molar agenesis in both males and females. For both sexes, the incidence of bilateral third molar agenesis in Group A was significantly higher than in Group B (*p* < 0.001). In males, the incidence of bilateral third molar agenesis in Group A was 2.9 times higher than in Group B; in females, it was 3.3 times higher. In Group B, similar to Group A, there was no sexual dimorphism in the prevalence of bilateral agenesis (*p* = 0.932, Table 3).

In Group B, the most common agenesis pattern was the agenesis of all third molars in both males and females (6.8% and 4.7%, respectively), followed by unilateral agenesis of the lower left third molar in males (3%) and bilateral agenesis of the lower third molars in females (3.5%) (Table 2). Similar to Group A, there was no sexual dimorphism in Group B in the bilateral versus the unilateral occurrence of the most common patterns (Table 2).

In Group B, as in Group A, there was no sexual dimorphism on the total number of missing third molars per individual, when distinct groups were formed based on agenesis severity (0, 1, 2, 3, or 4 missing molars, Figure 2, Appendix A). In all individual categories, Group A tended to have a higher third molar agenesis prevalence than Group B, although this did not consistently reach the level of significance (*p* < 0.01, Figure 2, Appendix A).

## 4. Discussion

The primary outcome of the current study was to investigate sexual dimorphism in the patterns and the severity of third molar agenesis in a large modern European sample with and without agenesis of other teeth. No sexual dimorphism in the incidence of third molar agenesis was observed within groups (neither with nor without agenesis of teeth other than the third molars). Sexual dimorphism in the different patterns of third molar agenesis, as well as in symmetry, was also investigated, and no difference was detected in both tested groups. Our findings are in accordance with previous studies that did not identify sexual dimorphism in the significant associations between the number of teeth with craniofacial size and shape in modern humans [10,11,12].

To our knowledge, the available evidence on sexual dimorphism in the patterns and the severity of third molar agenesis is scarce, since we were not able to identify any previous study investigating this topic. A meta-analysis performed on studies with different geographic backgrounds reported that women were 14% more susceptible to show at least one third molar agenesis than men, and that maxillary third molar agenesis occurred 36% more often than mandibular agenesis [1]. These findings are in contrast to our results, and there are several factors that could explain these differences. One important factor is geographic variation. The meta-analysis included various populations and the heterogeneity of the specific findings was high (50%), whereas no clear conclusion could be drawn when considering studies in only European populations (nine studies found higher prevalence in males and nine in females). Another important confounding factor that was not addressed in the meta-analysis was the presence of agenesis of teeth other than the third molars in the samples of the included individual studies [4]. No such eligibility criterion was applied. The reported occurrence of third molar agenesis in Europeans was 21.6%, while our study found only 11.3% of isolated third molar agenesis in males and 12.4% in females. The incidence of third molar agenesis was much higher in our study in the presence of concurrent agenesis of other teeth (48.9% in males and 52.4% in females). This points in the same direction of previous studies of different designs that showed an increased prevalence of agenesis of other teeth in individuals with third molar agenesis [16,17]. Another explanation for the inconsistency could be that the studies analyzed in the meta-analysis included younger individuals (11 years old and above) that might have developed third molars later, especially considering males. In our study, we included individuals older than 12.5 years to minimize this possibility [8,18,19,20,21]. According to previous studies, till 12.5 years, Demirjian’s stage A is observed in the third molars in most cases (95%). At this stage, the mineralization of the third molar cusps prevents a false-positive agenesis diagnosis in panoramic radiographs [8,18,19,20,21]. The 40-year-old upper limit was chosen to minimize misdiagnosed tooth agenesis of originally extracted teeth or teeth lost due to caries or other reasons. An additional factor that might have contributed to this inconsistency might be the inclusion of undiagnosed syndromic individuals in the studies considered in the meta-analysis, which could increase the incidence of third molar agenesis. Our data consisted of orthodontic records, which included a thorough medical and dental history, and the diagnosis was performed by trained orthodontists, reducing the possibility of misdiagnosis [22].

The large sample of 303 individuals per group, selected out of a total of around 10,000 patient files, allowed the investigation of the symmetry distribution of third molar agenesis, as well as potential differences in occurrences within quadrants. Such a large sample is presented for the first time in the literature, aiming to evaluate all the teeth in the dentition. No sexual dimorphism was found in the bilateral distribution of third molar agenesis in both groups (A, B). As reported previously [4], bilateral third molar agenesis was more often present than unilateral agenesis in both sexes when there was concurrent agenesis of teeth other than the third molars. On the contrary, there was no difference in the incidence of unilateral versus bilateral occurrence of isolated third molar agenesis. Bilateral third molar agenesis occurred 2.9 times more often in males and 3.33 times more often in females when agenesis was also present in the other teeth in the dentition compared to isolated third molar agenesis.

Regarding the sequence of the number of missing third molars per individual, no sexual dimorphism was found between and within the groups (A, B). Four third molars were most often missing in individuals who had additional agenesis of other teeth (Group A), whereas one third molar was most often missing in absence of agenesis of teeth other than third molars (Group B). The latter finding is in agreement to a relevant meta-analysis [1], which, however, did not consider sexual differences.

According to the aforementioned findings, tooth agenesis of other teeth in the dentition significantly affects third molar agenesis patterns. In particular, the probability of having four missing third molars increases considerably. In comparison with other tooth types, the third molars may be more vulnerable to genetic factors that cause tooth agenesis. Indeed, this is supported by the increased number of missing third molars in the agenesis sample (group A) compared to that expected by chance [4]. A recent study [15] that investigated 172 monozygotic twins and 112 dizygotic twins found that the formation of third molars was highly affected by additive genetic factors, which can also explain the increased third molar agenesis incidence in cases with agenesis of other teeth.

Genetic involvement in third molar agenesis was also revealed in a previous study, which found an association between third molar agenesis and reduced facial size, suggesting a common developmental mechanism that might be related to the evolutionary trend of reduced number of teeth and smaller faces in humans [12]. A similar finding was evident when considering the agenesis of any tooth type [11]. Furthermore, recent evidence suggests that individuals with tooth agenesis have different craniofacial features from those without agenesis [10], with the inclusion of the third molars in the analysis not affecting the outcomes [10]. The aforementioned studies did not identify any clear difference between sexes in the tested outcomes, which is in accordance with the findings of the present study regarding third molar agenesis patterns and also those of other teeth. Despite the increased prevalence of agenesis in female dentition compared to male dentition [2], which was not evident in our study concerning the third molars, in the presence of tooth agenesis, there was no sexual dimorphism in the agenesis patterns and severity for any tooth type, including the third molars. This indicates a strong genetic control of the developmental process of tooth formation, which affects both sexes in a similar manner when this process is disrupted, although such disruptions are more often observed in females regarding teeth other than third molars. The absence of any sexual dimorphism in third molar agenesis, including its incidence, is in contrast to the higher incidence in females observed for teeth other than the third molars, suggesting an absence of a strict sex-related genetic control of third molar development. This might be related to the evolutionary trend in humans leading to reduced number of molars, which, according to the current findings, affects both sexes similarly. Females also have smaller craniofacial size, and, thus, the higher incidence of tooth agenesis in them might be related to the evolutionary trend in humans towards a reduced number and size of teeth, as well as facial size [11,12,23,24,25].

From a clinical point of view, our study emphasizes the importance of accurate treatment planning in preadolescent patients with severe tooth agenesis. In these demanding cases, clinicians should take into consideration that the third molars have higher chances of being missing, often bilaterally. Thus, more complex multidisciplinary approaches might be required for successful management [26].

A secondary aim of the present study was to evaluate sexual dimorphism in the agenesis patterns of all teeth, including the third molars, in individuals with agenesis of teeth other than the third molars. The most common missing tooth was the lower second premolar for both sexes. When considering teeth other than the third molars, no sexual dimorphism was evident in our sample. This finding contrasts with a previous study, where sexual dimorphism was detected in tooth agenesis patterns, without considering third molars [5]. In that study, males showed higher agenesis incidence compared to females on the first premolars and the mandibular incisors (central and lateral) [5]. This inconsistency might be attributed to the fact that potentially undetected syndromic patients with full expression or subtypes might have been included in the sample of the latter study. The sample originated from a specialized treating center for tooth agenesis, which might have attracted more severe cases. In addition, more males with severe agenesis might have sought treatment there, compared to the general male population. Females seek treatment usually more often, even for less severe conditions [27]. Considering that our sample was selected from orthodontic patients, the latter confounder was probably not present. Moreover, the included individuals were thoroughly documented and followed over time; thus, the possibility that certain severe cases could represent undiagnosed syndromes was reduced. Finally, in that study, the average number of missing teeth, without considering the third molars, was 5.5 per individual, whereas it was 2.7 per individual in our study. Any of the aforementioned confounders could have been responsible for this contradictory finding.

In another recent study, sexual dimorphism was also found in Japanese patients with non-syndromic oligodontia [6]. In that study, without considering the third molars, the number of missing teeth per individual ranged from 6 to 19, while in our study, only 27 cases presented with 6 or more missing teeth (8.9% of the agenesis sample). Thus, this inconsistency could be attributed to the large differences in agenesis severity. The different geographical origin of the samples might have also affected the outcomes [1,2]. In agreement with our study, another study in Japanese individuals [14] tested sexual dimorphism in teeth other than the third molars and found no differences between sexes. However, the strict selection criteria applied in that study limited the generalizability of the findings. The study included individuals that had either agenesis of both mandibular third molars or of no third molar.

One limitation of our study can be the inclusion of orthodontic patients, which may vary in the severity of tooth agenesis incidence compared to the general population. According to several previous studies, tooth agenesis and malocclusion are endemic in recent years and tooth agenesis characteristics in orthodontic patients are not expected to vary greatly from those of the normal population [1,3]. Finally, as mentioned before, tooth agenesis can vary in severity and pattern depending on the geographical area or ancestry. To minimize possible confounding, only white European subjects were included in this study since they were overrepresented in the searched archives. Therefore, the present findings need to be confirmed in other populations.

## 5. Conclusions

This study showed no sexual dimorphism in the patterns or the severity of third molar agenesis. Both sexes showed significantly higher third molar agenesis, as well as bilateral occurrence, in the presence of agenesis of teeth other than the third molars compared to isolated third molar agenesis. The most common third molar agenesis pattern was the agenesis of all third molars for both groups and sexes. This indicates that the third molars might be more vulnerable to genetic or epigenetic factors involved in the agenesis of other teeth and they are often affected more globally. No sexual dimorphism was found in single tooth agenesis regarding all tooth types, including the third molars. This indicates a strong genetic control of the developmental process of tooth formation, which affects both sexes in a similar manner when this process is disrupted. Regarding the incidence of third molar agenesis, there is also no sex-related effect, in contrast to the increased female occurrence in teeth other than the third molars. Overall, this could be associated with the evolutionary trend in humans towards a reduced number of molar teeth, which seems to show no sex-related differences, but is consistent with the higher susceptibility of third molars to agenesis.

In preadolescent patients with multiple tooth agenesis, the higher probability of additional third molar agenesis should be considered during dental treatment planning, equally for both sexes.

## Figures and Tables

**Figure 1 biology-11-01725-f001:**
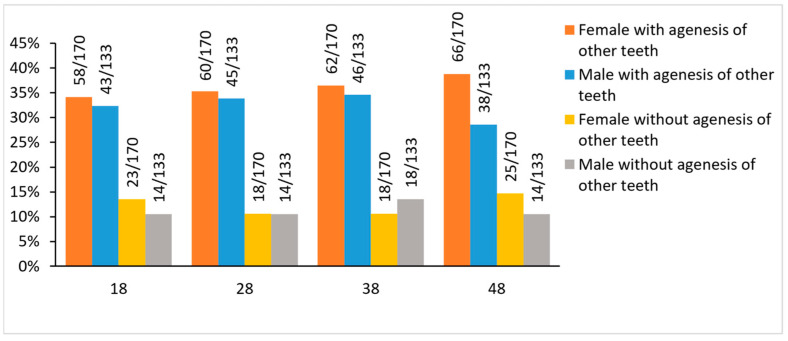
Third molar agenesis with and without agenesis of other teeth.

**Figure 2 biology-11-01725-f002:**
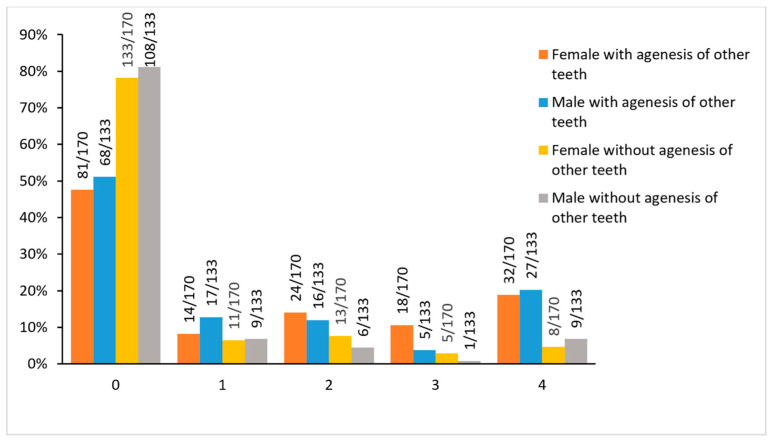
The total number of missing third molars with and without agenesis of other teeth.

**Figure 3 biology-11-01725-f003:**
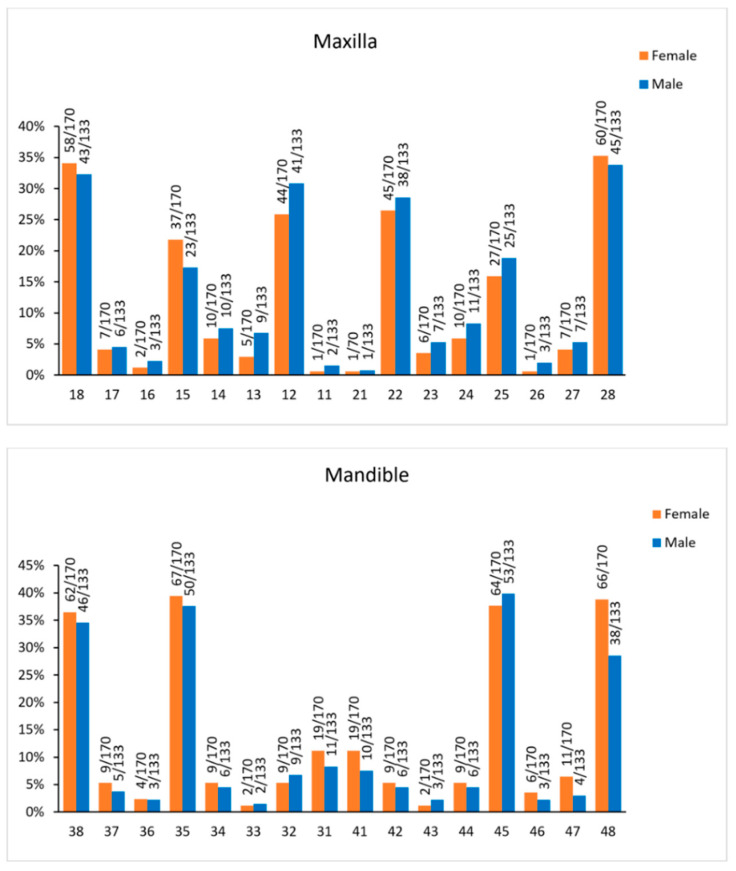
The number of missing teeth when considering all the teeth in the dentition including the third molars.

**Table 1 biology-11-01725-t001:** Distribution of the symmetry patterns of third molar agenesis per jaw in males and females with agenesis of teeth other than the third molars, overall and for each quadrant.

Gender	Present Bilaterally (%)	Missing Right Side (%)	Missing Left Side (%)	Missing Unilaterally (%)	Missing Bilaterally (%)	All Patients with Agenesis	Average Missing per Patient	All Missing Third Molars
**Maxilla**								
**Male**	83/133 (62.4%)	5/133 (3.8%)	7/133 (5.3%)	12/133 (9%)	38/133 (28.6%)	50/133 (37.6%)	0.66 ± 0.90	88/266 (33.08%)
**Female**	102/170 (60%)	8/170 (4.7%)	10/170 (5.9%)	18/170 (10.6%)	50/170 (29.4%)	68/170 (40%)	0.69 ± 0.90	118/340 (34.71%)
***p*-value**	0.669 ^1^	0.427 ^3^	1.000 ^3^	0.701 ^3^	0.872 ^1^	0.669 ^1^	0.725 ^2^	0.6751 ^1^
**Mandible**								
**Male**	85/133 (63.9%)	2/133 (1.5%)	10/133 (7.5%)	12/133 (9%)	36/133 (27.1%)	48/133 (36.1%)	0.63 ± 0.88	84/266 (31.58%)
**Female**	94/170 (55.3%)	14/170 (8.2%)	10/170 (5.9%)	24/170 (14.1%)	52/170 (30.6%)	76/170 (44.7%)	0.75 ± 0.90	128/340 (37.65%)
***p*-value**	0.158 ^1^	0.009 ^3^	0.643 ^3^	0.211 ^3^	0.529 ^1^	0.141 ^1^	0.198 ^2^	0.120 ^1^

^1^ Chi-square test, ^2^ Mann–Whitney test, ^3^ Fisher’s exact test (adjusted level of significance *p* < 0.006).

**Table 2 biology-11-01725-t002:** The most common third molar agenesis patterns in individuals with and without agenesis of teeth other than the third molars.

Gender	Most Common Patterns	Frequency (%)	Third Molar Agenesis Patterns	
			Bilateral ^1^	Unilateral ^1^
	With other teeth agenesis			
**Females**	1	32/170 (18.82%)	18, 28, 38, 48	-
	2	11/170 (6.47%)	38, 48	-
	3	9/170 (5.30%)	18, 28	-
	4–6	6/170 (3.50%) each	18, 28, 48 28, 38, 48	38
**Males**	1	27/133 (20.30%)	18, 28, 38, 48	-
	2	8/133 (6.01%)	38, 48	-
	3–4	7/133 (5.26%) each	18, 28	38
	5-6	5/133 (3.76%) each	-	18 28
	Without other teeth agenesis			
**Females**	1	8/170 (4.70%)	18, 28, 38, 48	-
	2	6/170 (3.52%)	38, 48	-
	3	5/170 (2.94%)	-	18, 48
	4–5	3/170 (1.76%) each	-	28 48
**Males**	1	9/133 (6.76%)	18, 28, 38, 48	-
	2	4/133 (3.00%)	-	38
	3	3/133 (2.25%)	38, 48	-
	4–6	2/133 (1.50%) each	18, 28	18 28

^1^ Within cells, each row represents a single pattern.

**Table 3 biology-11-01725-t003:** Incidence of bilateral versus unilateral third molar agenesis in individuals with and without agenesis of teeth other than the third molars.

	Females			Males			
	Bilateral	Unilateral	*p*-value ^1^	Bilateral	Unilateral	*p*-value ^1^	*p*-value ^2^
**With other teeth agenesis**	70/170 (41.18%)	19/170 (11.18%)	<0.001	47/133 (35.34%)	18/133 (13.53%)	<0.001	0.300
**Without other teeth agenesis**	21/170 (12.35%)	16/170 (9.41%)	0.383	16/133 (12.03%)	10/133 (7.52%)	0.215	0.932
** *p* ** **-value ^3^**	<0.001	0.592	-	<0.001	0.109	-	-

^1^ Bilateral versus unilateral third molar agenesis, ^2^ Sexual dimorphism in bilateral agenesis of third molars with and without agenesis of other teeth, ^3^ Differences in third molar agenesis between individuals with versus without agenesis of teeth other than the third molars.

**Table 4 biology-11-01725-t004:** The most common tooth agenesis patterns in individuals with agenesis of teeth other than the third molars when considering all the teeth in the dentition.

Gender	Most Common Patterns	Frequency (%)	Tooth Agenesis Patterns	
			Bilateral ^1^	Unilateral ^1^
**Entire dentition**				
**Females**	1	12/170 (7.06%)	12, 22	-
	2	10/170 (5.88%)	-	35
	3	8/170 (4.70%)	35, 45	
	4-6	7/170 (4.12%) each	-	12 22 45
**Males**	1	10/133 (7.52%)	12, 22	-
	2	9/133 (6.77%)	-	45
	3	8/133 (6.01%)	-	22
	4-5	6/133 (4.51%) each	35, 45	35
**Maxilla**				
**Females**	1	19/170 (11.18%)	18, 28	-
	2	16/170 (9.41%)	12, 22	-
	3–5	8/170 (4.71%) each	18, 15, 25, 28	12 22
**Males**	1	15/133 (11.28%)	12, 22	-
	2	11/133 (8.27%)	18, 28	-
	3	9/133 (6.77%%)	-	22
	4	8/133 (6.01%)	-	12
	5-7	4/133 (3.00%) each	12, 22, 18, 28	28 12, 18, 28
**Mandible**				
**Females**	1	17/170 (10.00%)	35, 45	-
	2	16/170 (9.41%)	-	35
	3	10/170 (5.88%)	38, 35, 48, 45	-
	4	8/170 (4.70%)	-	45
	5	7/170 (4.12%)	38, 48	-
**Males**	1	13/133 (9.77%)	35, 45	-
	2	12/133 (9.02%)	-	45
	3	10/133 (7.52%)	38, 35, 48, 45	-
	4	9/133 (6.77%)	38, 48	-
	5	8/133 (6.01%)	-	35

^1^ Within cells, each row represents a single pattern.

**Table 5 biology-11-01725-t005:** Distribution of the symmetry patterns of third molar agenesis per jaw in males and females without agenesis of teeth other than the third molars, overall and for each quadrant.

Gender	Present Bilaterally (%)	Missing Right Side (%)	Missing Left Side (%)	Missing Unilaterally (%)	Missing Bilaterally (%)	All Patients with Agenesis	Average Missing per Patient	All Missing Third Molars
**Maxilla**								
**Male**	116/133 (87.1%)	3/133 (2.3%)	3/133 (2.3%)	6/133 (4.6%)	11/133 (8.3%)	17/133 (12.9%)	0.21 ± 0.58	28/266 (10.53%)
**Female**	144/170 (84.7%)	8/170 (4.7%)	3/170 (1.8%)	11/170 (6.5%)	15/170 (8.8%)	26/170 (15.3%)	0.24 ± 0.66	41/340 (12.06%)
***p*-value**	0.537 ^1^	0.358 ^3^	1.000 ^3^	0.616 ^3^	1.000 ^3^	0.534 ^1^	0.558 ^2^	0.555 ^1^
**Mandible**								
**Male**	114/133 (85.7%)	1/133 (0.8%)	5/133 (3.8%)	6/133 (4.6%)	13/133 (9.8%)	19/133 (14.4%)	0.24 ± 0.62	32/266 (12.03%)
**Female**	141/170 (82.9%)	11/170 (6.5%)	4/170 (2.4%)	15/170 (8.9%)	14/170 (8.2%)	29/170 (17.1%)	0.25± 0.60	43/340 (12.65%)
***p*-value**	0.011 ^1^	0.015 ^3^	0.512 ^3^	0.174 ^3^	0.687 ^3^	0.618 ^1^	0.591 ^2^	0.818 ^1^

^1^ Chi-square test, ^2^ Mann–Whitney test, ^3^ Fisher’s exact test, (adjusted level of significance *p* < 0.006).

## Data Availability

All data are available in the main text or in the Appendix A. The protocols and datasets generated and/or analyzed in the current study are available from the corresponding author upon reasonable request.

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
