# Peer review of "Sexual Dimorphism in Third Molar Agenesis in Humans with and without Agenesis of Other Teeth"

_biology, 2022, doi:10.3390/biology11121725_

Round 1
Reviewer 1 Report
Dear authors, thank you for this interesting piece of research. This is a well designed and structured study.
Reviewer 2 Report
Nice presentation, well documented, widely discussed.
Reviewer 3 Report
This article deals with a very interesting topic. It is well written and the subject is discussed in a complete way"
Reviewer 4 Report
Dear authors,
I have reviewed your manuscript and i want to congratulate you for your work!
The manuscript is very well written, the introduction provides sufficient informations regarding the topic. I appreciate that the study methodology is clear and the inclusion and exclusion criteria are presented in detail.
I appreciate the data analysis which is without any errors.
Finally, the discussion section provides a comprehensive comparison between the study results and the scientific literature informations.
Reviewer 5 Report
hello
interesting paper, but somehow I dont understand it
1 - how come teeth is even compared to cranio-facial area malformation? (line 42-43). Its ridicolus since the term cranio-facial is way more sisious than a tooth! Perhaps Authors wanted to mention dento-alveolar discrepancies?.....
2- term agenesis and what Authors really are highlighting and evaluating should be clearly described - and its misisng
3- line 50-51 syndromic and non-syndromic? syndromic with what? other realted craniofacial deformities, clefts, underdeveloped tissues, OMENS? what? its unclear
4- line 71 concering teeth, wisdom teeth should be at the begining of the paragraph, as a basic introduction towards the problem
5- line 82-85, how do teeth impact on the cranial bones? perhaps maxillary and mandibular bones development is more precise? because of embryologic origins the bone from calvarian region/skull bones have way more different etiology than facial bones, sam as Tessier studies on crnaio-facial clefts indicates.
6- after reading the introduction I still dont know how do Authors describe agenesis and how do they define it in their study or similar cited in the introduction
7- line 113- european white or caucasians? nationality, habitation and other factors impact greately on teeth formation, especialy geographical differences
8 - part 2.2 data extraction- how do Authors verify so called "european white" from a single x-rays? or dental files?
9-metodology and material composition is unclear. study inclusion criteria are not well written, exlusion criteria from the study are not known
10- the average age of patients is a mystery - min/max age, occlusion and data related with tooth presence is unclear
11- the method on how Authors evaluated the dentition in different age groups based on the gathered material is unclear
12-how do Authors explain the data from a radiograph /panx/ from a patients aged 18 and 35, 70? all can have differet dentition and needs for dental treatement. No standrisation is described in the study
13-part 3.2 - I dont understand its sense
14- line 326- Our study... in young patients with sever tooth agenesis - Young is 5, 15 and 10 years of age, what about young adults? whats the patients age and what exactly crnaio-facial abormalities were evaluated? Cruzon, Syndrome, Apert, Pfeifer, other?....
15- line 362- basic limitations - poorly written and donest correlated with the study
I'm sorry to stay but In my opinion, from both a surgeon and past lecturer from human anatomy perspective, this study is far not suitable for publication. There is a big difference between craniofacial deformities, syndromic and non-syndromic patients and a simple teeth agenesis. This paper is not suitable for publication.
Round 2
Reviewer 5 Report
Hello
Thank you for the corrections
So far the article brings a little bit more sense to my understandings, however its still not something I would want to read. Mayby perhaps of my daily surgical routine Im more attached to the terms anomalies, craniofacial, skull and bones, rather then teeth. Its an interesting topic but for general dentists and orthodontists, who are mostly focusing on dental anomalies